# Did Mindful People Do Better during the COVID-19 Pandemic? Mindfulness Is Associated with Well-Being and Compliance with Prophylactic Measures

**DOI:** 10.3390/ijerph19095051

**Published:** 2022-04-21

**Authors:** Xinyue Wen, Ismaël Rafaï, Sébastien Duchêne, Marc Willinger

**Affiliations:** 1CEE-M, Université de Montpellier, CNRS, INRAE, Institut Agro, 34000 Montpellier, France; xinyue.wen@tsm-education.fr (X.W.); ismael.rafai@gmail.com (I.R.); marc.willinger@umontpellier.fr (M.W.); 2Toulouse School of Management, Université Toulouse 1 Capitole, 31000 Toulouse, France; 3GREDEG, CNRS, Université Côte d’Azur, 06000 Nice, France

**Keywords:** COVID-19, mindfulness, well-being, compliance

## Abstract

This paper investigates the relationship between mindfulness and well-being within the context of compliance with prophylactic measures in the time of COVID-19. We conducted a large-scale survey among a representative sample of the French population. We measured mindfulness, using the Mindful Attention Awareness Scale, and the extent to which respondents were impacted by COVID-19 in terms of their mood and quality of sleep, as well as how they complied with prophylactic measures. Our results suggest that more mindful individuals were less negatively impacted by COVID-19 with regard to their sleep and mood. Concerning the prophylactic measures, we obtained mixed results: more mindful participants were more likely to respect lockdowns, physical distancing and to cough in their sleeves, but did not wash their hands, wear masks or avoid touching their face more often than less mindful individuals.

## 1. Introduction

Since the outbreak of COVID-19 at the end of 2019, human beings have been confronted with an unprecedented health crisis. Physically, infected patients suffer from dry coughs and a fever along with possible acute respiratory distress syndrome [1], which can ultimately lead to death from multiple organ failure or complications from chronic serious illness [2]. Patients affected by COVID-19, individuals with mental disorders and healthcare workers, as well as the general population, experience different degrees of anxiety, stress, depression and insomnia [3,4,5], which often lead to increased addictive behaviors [6,7]. While several vaccines have become available to combat this pandemic, the negative impacts of COVID-19 on mental health are very latent and a timely response from psychological professionals is needed.

### 1.1. Mindfulness as a Protective Factor during the COVID-19 Crisis

Based on recent literature reviews [8,9], we hypothesized that mindfulness could be a protective factor against the impacts of COVID-19 on mental health. Mindfulness stems from an ancient Buddhist practice and an individual’s level of mindfulness can be improved using various meditation techniques [10]. Despite its religious origin, since its introduction into Western science, mindfulness has been adopted as a secular way to increase awareness and manage emotions. Therefore, in this context, mindfulness refers to the state or trait of an individual [11] and is defined as “paying attention in a particular way: on purpose, in the present moment, and nonjudgmentally” ([12], p. 4) or an awareness of current experience that is approached with curiosity, openness and acceptance [13].

In this paper, we investigate two potential channels through which mindfulness and health behaviors are related with respect to COVID-19 exposure: a direct channel, according to which mindfulness is associated with the lesser degradation of an individual’s mental health (e.g., sleep quality and mood), and an indirect channel, through which mindfulness is related to improved compliance with prophylactic measures and mediated by higher pro-sociability.

Mindfulness-based interventions, such as mindfulness-based stress reduction (MBSR) [12], have been proven to be beneficial for health in general [14] and, in particular: preventing mental disorders, including anxiety, stress and depression [15,16,17]; enhancing physical health [18], including improved immune functions and reduced blood pressure and cortisol levels [19]; improving psychological well-being, including higher self-evaluation and life satisfaction [20]; and better cognitive functioning, including improvements in attentional functions, cognitive flexibility [21] and working memory capacity [22,23]. Based on these findings, we hypothesized that mindfulness could make people less likely to be negatively affected by the COVID-19 pandemic, particularly with regard to sleep quality and mood. We selected these two well-being markers because they are involved in two vicious circles that are likely to aggravate health problems.

### 1.2. Relationship between Mindfulness and Sleep Quality and Mood

Several recent studies have suggested an association between sleep quality and states of crisis, such as the COVID-19 outbreak [24,25,26]. Based on a systematic review and a meta-analysis (N = 221,970), ref. [27] found that sleep disorders have been the most prevalent mental and psychological health problems worldwide during the COVID-19 pandemic, especially among physicians and nurses. This poses a problem that is particularly worrying as sleep quality affects perceived stress and depression [28], which in turn lead to sleep disorders and hence, a vicious circle. In addition, bad sleep quality is often associated with bad mood, although there are also independent causes for bad mood. Moods occur in a vicious (virtuous) circle that is similar to that of bad (good) sleep quality. Recent systematic reviews and meta-analyses have highlighted the relationship between mood state and illness. Ref. [28] established that susceptibility to contracting COVID-19 is associated with preexisting mood disorders and that the illness severity is associated with both preexisting and subsequent mood disorders, as well as sleep disturbance. Similarly, ref. [29] suggested that individuals with preexisting mood disorders are at a higher risk of hospitalization and death due to COVID-19. Given the above evidence regarding the vicious/virtuous circles that are associated with sleep quality and mood, we targeted these two dimensions as potential vectors of influence for mindfulness.

In their systematic literature review, ref. [30] reported three studies that show the positive impact of MBSR on sleep quality [31]. In addition, ref. [32] showed that the trait of mindfulness affects sleep quality through the mediation of negative emotions. However, ref. [33] reported a weak or no effect on university students in their systematic review and meta-analysis. Furthermore, the impact of mindfulness on sleep quality has not been documented for a general population that has been exposed to a pandemic outbreak. With respect to mood, the literature survey of [34] showed that mindfulness is an effective strategy for the treatment of mood disorders and anxiety, a conclusion that was recently confirmed by [35,36], even in the case of brief mindfulness sessions. Gratitude seems to be a major mediator for the impact of mindfulness on mood [37,38]. Related to the COVID-19 pandemic, many scholars have advocated mindfulness as a means to prevent mental health issues [39,40,41,42].

We expected that both sleep quality and mood could represent relevant markers for COVID-19-related impacts on well-being. Specifically, we stated our first hypothesis as follows:

**Hypothesis** **1** **(H1).**
*Mindfulness is associated with reduced adverse psychological effects of COVID-19, as measured by sleep quality and mood.*


### 1.3. Relationship between Mindfulness and Compliance with Prophylactic Measures

The second channel through which mindfulness is related to health behavior is mediated through pro-sociability: ref. [43] found higher levels of altruism in mindful individuals; ref. [44] reported enhanced pro-social behaviors with compassionate feelings; and [45] documented higher cooperativeness. The systematic review and meta-analysis by [46] provide additional support for the positive influence of mindfulness on pro-sociability, empathy and compassion, both in terms of stated outcomes and observable outcomes, such as helping others. As mindfulness seems to promote positive health behavior, it could possibly lead to increased compliance with barrier gestures (i.e., protective measures, such as washing hands, coughing into your sleeve, avoiding touching faces and wearing masks) during the COVID-19 crisis. These measures were proposed by the World Health Organization to reduce the chance of infection. Ref. [47] found that higher pro-sociability (measured based on GPS items ([48], 2018)) predicts improved health behavior, including compliance with barrier gestures (e.g., coughing and sneezing in elbow, washing hands, not touching face, etc.), self-isolation in the case of contamination, avoiding social contact, buying a mask and many others.

We stated our second hypothesis with respect to compliance with sanitary measures, particularly applying barrier gestures (washing hands, coughing into your sleeve, not touching faces and wearing masks) and respecting social distancing measures (physical distancing and lockdowns), as follows:

**Hypothesis** **2** **(H2).**
*Mindfulness is associated with higher compliance with prophylactic measures, including barrier gestures and social distancing measures.*


### 1.4. Previous Evidence Collected during the COVID-19 Crisis

Based on a large literature survey, ref. [49] recommended mindfulness and the practice of meditation as a means to mitigate the adverse effects of the COVID-19 crisis on mental health. Moreover, ref. [50] suggested the practice of yoga as a moderating factor. Several papers have investigated the role of mindfulness in the COVID-19 crisis more closely. Most of them have established supportive evidence for H1: that mindfulness-based training or mindfulness practice is associated with reduced negative impacts of COVID-19 on well-being [40,41,51,52,53]. In some of these studies, the impact of mindfulness training on mental health was observed directly, either in the general population [53] or in a targeted group, such as students [40,52]. Other studies [41,51] relied on the measurement of mindfulness awareness using the Mindfulness Attention Awareness Scale (MAAS) questionnaire of [54] (more descriptions are provided in Section 2). The limitation of the above studies is that they are not representative of a population and therefore, cannot offer a relevant basis for designing public policies that are based on mindfulness. In contrast, our study proposed an assessment of the impact of mindfulness awareness on a representative sample of a national population (France).

As for the impacts of mindfulness on sleep quality during COVID-19, the available evidence, which was mostly collected during lockdown periods, is mixed. Some studies have reported a positive effect on sleep quality during the COVID-19 crisis among college students [55], physicians and advanced practice providers [56], nurses [57] and the general population [58]. In contrast, [59] only found a positive buffering effect on sleep duration but no effect on sleep quality, both in a Chinese sample and a British sample. However, ref. [60] observed that mindfulness training had a beneficial effect on sleep quality when the training was delivered before, but not during, the outbreak. All of the above-mentioned studies were based on small samples (most involved less than a hundred individuals). Therefore, more evidence that is based on large representative samples is needed before a sound conclusion can be reached.

A handful of papers have investigated the issue of a potential link between mindfulness and compliance with preventive COVID-19 measures. Ref. [61] observed that mindfulness is positively related to compliance with physical distancing. Ref. [62] found that behavioral and mindful emotion regulation skills are related to greater adherence to COVID-19 restriction measures. Ref. [63] observed that contemplative practice behaviors are associated with lockdown compliance. More generally, cognitive attitudes seem to be a strong predictor of compliance, according to [64].

### 1.5. Our Contribution

To test our hypotheses, we conducted a large web survey on a representative sample of the French population, in which we measured (i) mindfulness (using the MAAS questionnaire), (ii) the adverse effects of COVID-19 on sleep quality and mood (using a self-declared questionnaire) and (iii) self-declared compliance with prophylactic measures, including barrier gestures (washing hands, coughing into your sleeve, not touching faces and wearing masks) and social distancing measures (physical distancing and lockdowns). We applied ordered probit models to regress well-being and compliant behaviors onto mindfulness, using some basic sociodemographic variables as controls, and then computed the marginal effects of mindfulness on each dependent variable.

Our analysis suggested that mindfulness served as a buffer against the negative effects of COVID-19 on sleep quality and mood. As for the relationship between mindfulness and prophylactic measures, there was a significantly positive association between mindfulness and social distancing measures (physical distancing and lockdown compliance), but the correlation between mindfulness and barrier gestures (washing hands, coughing into your sleeve, not touching faces and wearing masks) were mixed: only coughing into your sleeve was positively and significantly associated with mindfulness; all of the others were not significant (although positive).

Our paper contributes both to the literature on mindfulness and the literature on COVID-19. There are three main contributions of our study. Firstly, it was the first study to assess the MAAS scores of a large nationally representative sample, which provides a good illustration of the mindfulness level of the French population. However, our survey was conducted before the end of the first national lockdown in France; thus, the national mindfulness level that was captured could represent a unique feature compared to that we would have obtained if the survey were administered at another time. Measuring the MAAS of such a large and representative sample of a population provided a much broader scope for the results of our study compared to those that have previously been presented by small samples, non-representative recruitments, student samples or very specific populations. Our research also contributes to the validity of the MAAS and the robustness of the survey and the corresponding experimental approaches by comparing our results to those in the existing literature. Secondly, our study identified a correlation between mindfulness and the impacts of COVID-19 on sleep quality and mood for a representative sample of the French population. Thirdly, it analyzed the association between mindfulness and six compliant behaviors and was, so far, the first paper to embrace this number of preventive measures in a single study on mindfulness. Furthermore, those measures were not exclusive to France, but have been commonly implemented in other countries, which makes the results of our study transferable to other populations. Indeed, we intentionally selected four of the main barrier gestures that were proposed by the World Health Organization and the French government, i.e., (a) washing your hands, (b) coughing into your sleeve, (c) avoiding touching your face and (d) wearing a mask. To these, we added two of the main social distancing measures: physical distancing and another more broadly regarding compliance with the lockdown guidelines. Beyond their relevance to the anti-COVID-19 strategies that have been implemented by governments, we selected these measures because their relationship with mindfulness might differ considerably, with some of them potentially involving different mediators (such as pro-sociability, risk aversion, cooperativeness or empathy).

Finally, if our hypotheses and correlations were validated by this large representative group of French participants, they could provide a starting point for the design of public policies that are aimed at improving individual well-being and increasing compliance with prophylactic measures during times of crises or severe stress periods through the practice of meditation, both in France and also more widely.

The rest of this paper is organized as follows. The next section presents our methodology. The results are presented in Section 3 and discussed further in Section 4.

## 2. Method

### 2.1. Data and Survey

The survey was part of a comprehensive project in which data were also collected for other research projects and were ultimately presented in other papers [65,66,67,68,69]. For example, refs. [65,66] aimed to determine the common preferences of the French population for lockdown characteristics through a discrete choice experiment. Ref. [67] investigated the effectiveness of a nudge toward compliance during a future hypothetical lockdown in a controlled experiment. Ref. [68] investigated the impact of economic preferences (in social, time and risk dimensions) on compliance with prophylactic measures. Ref. [69] investigate the relation between age and economic preferences.

The survey started on 4 May 2020 and lasted until 16 May 2020. The first French national lockdown lasted from 17 March to 10 May 2020. Using the quota method, the survey institute *Viavoice* (http://www.institut-viavoice.com/, accessed on 15 April 2022) telephoned more than 7500 people at the end of March 2020 and asked them to participate in an online study, which included the present survey. Those who declined the invitation to participate were replaced (as far as possible) by other people with similar profiles. In total, 5331 persons agreed to participate and received a link to complete the survey using a dedicated server that was managed by the research team. Those individuals were representative of the French population in terms of their gender, age, professional and social categories (PCS, INSEE definitions), geographical area (UDA-9) and the size of their urban unit (INSEE definition).

Ultimately, 1154 individuals completed the survey and signed the informed consent form (response rate of 21.6%). In terms of gender, age and geographical area, the surveyed sample was a good representation of the whole French population, with a slight over-representation of high social status males who were older than 50. Table 1 presents the statistics of the INSEE (French National Institute of Statistics and Economic Studies) official national panel for the 5331 people who agreed to participate in the survey and the 1154 participants who completed it.

### 2.2. Mindfulness Attention Awareness Scale

We chose the Mindfulness Attention Awareness Scale (MAAS; [54]) to gauge the mindfulness level of the French population. This scale, as its name suggests, measures the essential facets of mindfulness: attention and awareness. The respondents are asked to provide an answer for the 15 items of the questionnaire using a 6-point Likert scale, which rates the frequency of everyday experiences from 1 (almost always) to 6 (almost never). The final MAAS score is the mean of the answers to the 15 items. The higher the average score, the more the individual is considered to be mindful. The version we used, which was translated into French by ourselves, is available in Appendix A. We found a Cronbach’s alpha of 0.842, which was comparable to those found by other authors for different sample groups (ranging from 0.80 to 0.87; [54]).

### 2.3. Well-Being

In the survey, the impact of COVID-19 on well-being was measured through two aspects: quality of sleep and general mood. More precisely, we first asked respondents if the COVID-19 crisis had impacted their sleep and if yes, to what extent. Similar questions were asked regarding the impact of COVID-19 on their mood. The impacts of COVID-19 on sleep quality and mood were measured using a 5-point scale (0 = “Considerably deteriorated”, 1 = “Deteriorated”, 2 = “No impact”, 3 = “Improved”, 4 = “Considerably improved”), as depicted in Figure 1. The lower the measure, the worse the person had been affected by the COVID-19 pandemic. More precisely, we asked the following questions. **Q1** (all participants): “*Has the COVID-19 crisis affected your sleep?*” (Possible answers: “*No*”, “*Yes*”). **Q2** (only to participants who answered “*Yes*” to Q1): “*If yes, to which degree?*” (Possible answers: “*Considerably deteriorated*”, “*Deteriorated*”, “*Improved*”, “*Considerably improved*”). **Q3** (all participants): “*Has the COVID-19 crisis affected your mood?*” (Possible answers: “*No*”, “*Yes*”). **Q4** (only to participants who answered “*Yes*” to Q4): “*If yes, to which degree?*” (Possible answers: “*Considerably deteriorated*”, “*Deteriorated*”, “*Improved*”, “*Considerably improved*”). Answers to Q1 and Q2 (resp. Q3 and Q4) were merged to obtain a complete ordinal scale for the impacts on sleep quality (resp. mood). Participants who answered “*No*” to Q1 (resp. Q3) were considered as “*No impact*” and were integrated into the middle of the Q2 (resp. Q4) scale (in between “*Deteriorated*” and “*Improved*”). Since this covered all situations, only the analyses of the 5-point ordinal scales are presented in the following sections. Qualitatively similar results were obtained when the questions were analyzed separately, i.e., more mindful individuals were (i) less likely to be impacted in terms of the quality of their sleep and their mood and (ii) if they were impacted, they were less likely to be negatively impacted.

### 2.4. Compliant Behaviors

In our survey, questions were asked about two types of prophylactic measures. The first type was barrier gestures, including washing hands, coughing into your sleeve, not touching faces and wearing masks. The other type was social distancing measures, i.e., physical distancing and lockdowns. The detailed questionnaire is presented below. One thing to note is that wearing a mask was not yet obligatory before the end of the lockdown due to a supply shortage and some ambiguities about its effectiveness at the beginning of the pandemic in France. Hence, that question was in the simple future tense to investigate the future willingness of the respondents rather than their actual compliance.

Type 1: Barrier Gestures

“During the lockdown, when you went out of your home, did you respect the following recommendations?”

Wash your hands (1 = “Never”, 2 = “Sometimes”, 3 = “Often”, 4 = “Very Often”, NA = “I don’t know”)

Cough in your sleeves (1 = “Never”, 2 = “Sometimes”, 3 = “Often”, 4 = “Very Often”, NA = “I don’t know”) 

Avoid touching your face (1 = “Never”, 2 = “Sometimes”, 3 = “Often”, 4 = “Very Often”, NA = “I don’t know”)

Mask wearing: “As long as the virus circulates in France, when you go out in public places (stores, city center), will you wear a mask?” (1 = “Never”, 2 = “Rarely”, 3 = “Often”, 4 = “Always”, NA = “I don’t know/I never go out”)

Type 2: Social Distancing Measures

Physical distancing: “Respect a distance of at least one meter with other people.” (1 = “Never”, 2 = “Sometimes”, 3 = “Often”, 4 = “Very Often”, NA = “I don’t know”)

Lockdown: “Would you say that you strictly respect the lockdown governmental directive?” (Answer on a scale from 1 to 10, 1 = “Not at all” and 10 = “strictly”)

Participants that answered “I don’t know” were removed from the analysis.

### 2.5. Control Variables

As control variables, we collected sociodemographic characteristics (i.e., gender, age, monthly income and education level) and COVID-19-related characteristics, such as being a “vulnerable person” (whether the respondent suffers from a chronic illness that makes them vulnerable to the threat of COVID-19) and living conditions (whether the respondent lives with someone who is old or suffers from a health problem that makes that person vulnerable to threat of COVID-19). The control variables are described and summarized in Table 2.

Previous research supported the selection of the control variables. On one hand, studies have shown that sociodemographic factors have some influence on well-being and compliant behaviors. For example, ref. [70] demonstrated that healthy older respondents manifested significantly disturbed sleep relative to healthy younger respondents, according to both subjective and objective reports. Ref. [71] found that female respondents respected government rules and health precautions more than males during the initial spread of COVID-19. Ref. [72] found that those with a university diploma showed a higher level of compliance with prophylactic measures compared to those without a higher education, which suggests that education could be positively correlated with compliance.

In terms of monthly income, a higher level of comfort in the home lowered the cost of being locked down. As for vulnerable people and living conditions, those who were confronted with higher risks after infection were more willing to respect the prophylactic measures in order to reduce the possibility of getting infected. However, this conjecture should be tempered as there could be a large difference between objective and perceived risks, especially after the COVID-19 pandemic. Ref. [73] pointed out that many experimental studies have documented human insensitivity to mass tragedies with regard to attitudes toward risk [74]. Accordingly, this may be the case with the COVID-19 pandemic as well. When individuals are confronted with the millions of deaths worldwide that were related to this disease, it is possible that populations underestimate the risk of being infected and the risks that develop after infection. Thus, it could be that facing a higher risk does not necessarily lead to a greater willingness to comply with prophylactic measures to minimize the chance of becoming infected, even for objectively vulnerable people and those in vulnerable living conditions.

## 3. Results

Our data analysis proceeds as follows. We first present the distribution and descriptive statistics of the MAAS scores in Section 3.1. Then, the distribution of well-being (impact on sleep quality and mood) and prophylactic measures, as well as the correlations between the prophylactic measures, are shown in Section 3.2. Finally, we present the explanatory power of mindfulness for all of the dependent variables in Section 3.3.

### 3.1. Distribution and Descriptive Statistics of the MAAS Scores

The distribution and descriptive statistics of the MAAS scores are presented in Figure 2 and Table 3, respectively. For the distribution of the MAAS scores, the first quartile in our sample was at 3.87. Moreover, we can see clearly from Figure 2 that the MAAS score distribution of the French population was left skewed, which indicated that only a small proportion of the population achieved a low score on the MAAS.

Table 3 displays a comparison of some descriptive statistics of MAAS scores from other studies [54,75,76,77] whose sample was from the general population, not just teenagers, cancer patients or a specific type of population. However, unlike our sample, these samples were not representative of a national population in terms of gender and age. Furthermore, the subjects were commonly from one local community (except for [77] and the sample sizes were limited, while our subjects were from every department in metropolitan France and the sample size was up to 1154 (in our survey, every respondent replied to the MAAS questions so there were no missing observations concerning this measure). Thus, our study was the first to measure the mindfulness level of a representative population of a country using the MAAS.

From Table 3, we can see that the average MAAS score of the French population was quite high compared to that of people in other countries. However, we should be cautious in declaring that the French population is mindful in daily life because the data were collected during the pandemic. People might have been more aware of their bodily sensations than in normal times; thus, the average MAAS score in our study can only be interpreted as a measure of mindfulness during a pandemic.

### 3.2. Distribution and Correlation of Dependent Variables

#### 3.2.1. Well-Being

For sleep quality and mood, the percentage of people who were positively affected could be ignored as it was negligible. For those who were negatively impacted, 24.70% of the respondents suffered from sleep disturbance and 32.84% experienced bad mood. The percentage of people who were impacted by the health crisis in terms of their sleep quality and mood was moderate in France compared to other cross-sectional studies that were conducted up to August 2020 [78]. In addition, COVID-19 seemed to cause more trouble with mood than sleep quality. The distribution of the impacts of COVID-19 on sleep quality and mood is represented in Figure 3.

#### 3.2.2. Compliant Behaviors

Figure 4 presents the distribution of compliance with each prophylactic measure. Most respondents reported a high level of compliance with all of these measures with the exception of not touching faces, for which compliance was less unanimous.

Table 4 displays the pairwise Pearson correlation between the different measures. Overall, positive correlations existed among all of the prophylactic measures with the exception of the correlation coefficient between physical distancing and coughing into your sleeve, which was equal to 0.051 at the 10% significance level. All of the other values ranged from 0.108 to 0.239 at the 0.1% level of significance. In contrast to nearly all significant correlations, the Cronbach’s alpha of all of the measures equaled 0.495, which means that these measures were correlated but not redundant. As a result, a regression analysis was performed for each prophylactic measure in the following subsection.

### 3.3. Explanatory Power of Mindfulness

We first conducted ordered probit regression models (OP models) with well-being (i.e., sleep quality and mood) and compliant behaviors (all three types that were mentioned in Section 2) as dependent variables. The independent variables were the MAAS scores and the control variables, including the sociodemographic and COVID-19-related characteristics. Effects that were significant at a level of 5% or lower are discussed while those at a 10% significance are only reported for informative purposes. Furthermore, to better understand the explanatory power of mindfulness, the marginal effect in each regression model was then computed. All of the data analyses were performed using R.

#### 3.3.1. Well-Being

The MAAS score was positively correlated with the degree of impact on sleep quality ($Spearman^{\prime }s\;\rho =0.101;\;\;p=0.0006$) and with the degree of impact on mood ($Spearman^{\prime }s\;\rho =0.109;\;\;\;p=0.0002$). Table 5 displays similar results that were obtained with ordered probit regression models that predicted the effect of mindfulness on the degree of impact of COVID-19 on well-being: the MAAS score was positively correlated with sleep quality ($\beta_{MAAS}=0.1285;\;\;\;t=2.2629;\;\;\;p=0.0236$) and mood ($\beta_{MAAS}=0.1516;\;\;\;t=2.8889;\;\;\;p=0.0039$). This result supported H1, i.e., more mindful respondents were less likely to be negatively impacted by COVID-19.

Concerning the control variables, some were significantly related to impacts on well-being. Women reported being more impacted by the pandemic than men, especially regarding sleep quality (($\beta_{men}=0.3174;\;\;t=3.8902;\;\;p=0.0001$) for sleep quality and ($\beta_{men}=0.1230;\;\;\;t=1.6528;\;\;\;p=0.0984$) for mood). Younger respondents were more likely to be negatively affected in terms of their mood ($\beta_{age}=0.0103;\;\;\;t=4.4234;\;\;\;p<0.0001$) and sleep quality ($\beta_{age}=0.0060;\;\;t=2.3479;\;\;p=0.0189$). These significant correlations between gender, age and well-being were consistent with previous research on COVID 19. Ref. [79] showed that women and younger individuals reported higher levels of a sense of danger and distress symptoms during the COVID pandemic. In addition, ref. [52] reported that meditation could have a positive impact on some adolescents by gradually increasing their low levels of resilience during the pandemic. In addition, vulnerable people experienced significantly worse impacts on sleep quality ($\beta_{vulnerable}=-0.1667;\;\;t=-2.7375;\;\;\;p=0.0062$), but not significantly worse impacts on mood ($\beta_{vulnerable}=-0.0593;\;\;\;t=-1.0494;\;\;p=0.2940$). Monthly income, education level and living conditions were not significant.

The coefficients of the variables that were estimated by the ordered probit regression models did not reflect the effects of contributing factors for each level because there were five levels for each dependent variable (0 = “Considerably deteriorated”, 1 = “Deteriorated”, etc.). The marginal effect was subsequently calculated to reveal the change in the occurrence probability of each level of the dependent variable, which was associated with a unit increase in each independent variable, keeping all the other variables constant.

Since the focus of this article is the role of mindfulness, we only report the marginal effects of the MAAS scores on each level of sleep quality and mood in Table 6. We computed the marginal effect at the mean (MEM) and the average marginal effect (AME) of the MAAS scores. These two types of marginal effects yielded similar results, not only in terms of the significance of the coefficients but also their values. This also applied to the marginal effects of compliant behaviors that were reported in Section 3.3.2. To simplify the interpretation, we only discuss the AME in both subsections. MEM is the partial effect of the MAAS score on the change in the occurrence probability at an effect level, thereby keeping the other covariates at the mean. In contrast, the AME first estimates the partial effects of the MAAS score on the change in the occurrence probability at an effect level using the observed values for the other covariates and then averages the partial effect.

All AME values were significant at the conventional level of 5%, except that of sleep quality at levels 3 and 4, which were only significant at the 10% level. In particular, a higher level of mindfulness was associated with a lower probability of being negatively impacted by COVID-19, both in terms of sleep quality and mood. Specifically, a unit increase in the MAAS score was associated with a probability reduction in sleep being “Deteriorated” or “Considerably deteriorated” of 2.75% and 1.07%, respectively. With regard to mood, the corresponding marginal probability reductions in “Deteriorated” or “Considerably deteriorated” mood were estimated to be 4.40% and 0.89%, respectively. An increase in the MAAS score was also associated with an increase in the probability of not being affected by COVID-19 (“No impact”) in terms of sleep quality (+3.38%) and mood (+4.09%). The impact of the MAAS score on improvements in sleep quality and mood was negligible because there were so few subjects reporting improved sleep quality or mood. Lastly, the protective effect of mindfulness on mood was more prominent because the absolute values of its coefficients were greater than those of sleep quality, apart from level 0.

To conclude, mindfulness was related to a considerable reduction in the probability of negative impacts on both sleep quality and mood and is therefore considered to be a good buffer for the protection of individual well-being against health crises.

#### 3.3.2. Compliant Behaviors

Concerning the barrier gestures, the MAAS score was positively but not significantly correlated with Coughing in sleeves $(Spearman^{\prime }s\;\rho =0.0398;\;\;\;p=0.1835$) and was positively and significantly correlated with washing hands (Spearman’s ρ=0.109;   p=0.0002), not touching faces ($Spearman^{\prime }s\;\rho =0.0941;\;\;\;p=0.0016)$ and wearing masks ($Spearman^{\prime }s\;\rho =0.0841;\;\;\;p=0.0048$). Concerning social distancing measures, the MAAS score was positively and significantly correlated with physical distancing ($Spearman^{\prime }s\;\rho =0.1179;\;\;\;p<0.0001$) and compliance with lockdowns ($Spearman^{\prime }s\;\rho =0.0839;\;\;\;p=0.0044$). Table 7 shows the ordered probit regression model for the prophylactic measures. In general, the coefficients of the MAAS score for every measure were positive, which means that a higher MAAS score was associated with a higher reported level of compliance. However, only three out of the six measures were acceptable regarding significance: coughing into your sleeve and physical distancing were significant at the 5% level and lockdown compliance was significant at the 1% level. In other words, only one (Coughing in sleeves) of the four barrier gestures was significantly affected by the MAAS score, while the two measures of social distancing (physical distancing and lockdown compliance) were both significantly affected by the score. The marginal effects that are reported below (Table 7, Table 8 and Table 9) confirmed these findings.

The control variables showed similar results to the regression models for well-being. Gender and age were two predictors of compliant behaviors. All of the coefficients for gender were negative at the 0.1% level of significance, except for physical distancing at the 10% level. This consistent result showed that women were more compliant with prophylactic measures, in general, compared to men. These results could be related to some other gender effects that were observed during the COVID-19 pandemic. Ref. [7] found that during lockdown, women were more likely to fall into addictive behavior (losing control of their usual diet, using smartphones, etc.). More broadly, the emotional and behavioral responses that were generated by the pandemic appeared to vary widely by gender. For example, ref. [80] showed in a study of 3088 Chinese participants that women reported and experienced a higher level of psychological stress than men during the pandemic.

Age was a significant predictor of all of the measures, whereas its impact could be opposite in different measures. The old tended to pay less attention to washing hands and coughing into your sleeve than the young, while they respected all of the other measures more strictly, i.e., physical distancing, not touching faces, wearing masks and complying with lockdowns. Other control variables were not statistically significant, apart from two positive correlations: more educated people tended to conform more with coughing into your sleeve and vulnerable people were more likely to wear masks in public.

To better understand the explanatory power of mindfulness, we report the MEM and the AME of mindfulness on each measure in Table 8, Table 9 and Table 10. Since the MEM and the AME provided similar results, we only discuss the AME.

The AME values were only significant for compliance with coughing into your sleeve, physical distancing and lockdowns and not for the other variables, which confirmed the previous results that mindfulness was a more powerful predictor of distancing measures compared to barrier gestures.

For those three prophylactic measures, only the marginal coefficients for the highest levels (level 4 for coughing into your sleeve and physical distancing and level 10 for lockdown compliance) were positive. This feature indicated that a unit increase in the MAAS score was associated, on average, with an increase in the probability of maximal compliance with coughing into your sleeve, physical distancing and lockdowns of 5.73%, 3.49% and 5.65%, respectively, to the detriment of the lower levels of compliance. In particular, an additional unit in the MAAS score was associated with an estimated decrease in the probability of never coughing into sleeves or respecting physical distancing by 1.55% and 0.18%, respectively. The coefficients of lockdown compliance on lower levels were either non-significant (for levels 1 to 4, probably because of an insufficient number of observations) or too small to make a difference.

In brief, mindfulness was associated with compliance with distancing measures, but its relationship with barrier gestures was far less definite. Three of the four barrier gestures (i.e., washing hands, not touching faces and wearing masks) failed to satisfy the 5% conventional significance level, yet all of their coefficients were positive and the associated *p* values were lower than 10%.

## 4. Discussion

We investigated the predictive power of mindfulness on well-being and compliance with prophylactic measures against the COVID-19 disease during the first national lockdown in France. We hypothesized that (H1) higher levels of mindfulness could be associated with reduced adverse effects of COVID-19 on mood and sleep quality and that (H2) higher levels of mindfulness could be associated with higher compliance with the prophylactic measures against COVID-19.

While our findings unambiguously confirmed H1, the results were less clear with respect to H2. Indeed, we found that mindfulness was significantly related to higher compliance with some (physical distancing, lockdown and coughing into your sleeve) but not all of the measures (washing hands, not touching faces and wearing masks).

In our study, we found that mindfulness was associated with reduced negative effects of COVID-19 on mood and sleep quality. This finding suggested that mindfulness could be a helpful resource in overcoming difficult times, such as the current global health situation. Our results could be used to support the recommendations of [40,42,49,51] for example, who advocate mindfulness as a means to temper the adverse effects of the COVID-19 crisis on mental health. Mindfulness is the awareness of current inner and outer experiences that is approached with openness and acceptance [12,13], which induces disengagement from self-concern. When disturbing emotions occur, mindfulness allows people not to engage with those emotions and instead to directly focus on awareness itself, which is an effective measure for the management of emotions [81]. Therefore, our findings about the protective effects of mindfulness on mental health during the COVID-19 pandemic were consistent with the existing research [40,41,51,52,53].

Concerning the relationship between mindfulness and sleep quality, things are more intertwined. Previous literature has illustrated improved well-being on the psychological, physical and social levels [82] and decreased rumination [83] through the practice of mindfulness can together result in a better quality of sleep. While a lot of studies that were conducted before the pandemic have confirmed the association between mindfulness and sleep quality [84,85,86], studies that investigated this relationship during the pandemic, mostly during lockdown periods, have reported mixed evidence. Positive effects were reported by [55,56,57,58], but the findings of [59,60] were more puzzling. Our data clearly showed that mindfulness could help to reduce the negative impacts of COVID-19 on sleep quality. However, our measure of sleep quality was based on the self-assessment of the respondents, which could be biased. The same limitation applied to [59]. In contrast, ref. [60] used the 19-item Pittsburgh Sleep Quality Index to assess sleep quality, along with two other sleep variables (total sleep time and sleep onset latency), to obtain a comprehensive measure of sleep quality.

Overall, further investigations are necessary to better understand the relationship between mindfulness and sleep quality and to establish a causal relationship. Our study validated the positive correlation between the level of mindfulness and well-being, based on a large representative sample of the French population and thus, can be used to generalize many experimental studies or surveys that were previously performed on smaller samples. It also confirmed and extended the theoretical and experimental results that were obtained in situations in the absence of major stressful events to a severe pandemic situation with an exogenous shock of extremely high stress. The next steps that are related to this finding are to assess the causal link between meditation practice and well-being within the high-stress context that is related to the COVID-19 pandemic, to apply an experimental approach and to understand the key mechanisms (mediators) behind these findings. In addition, these results could pave the way for public policies that promote the practice of meditation for improving the well-being of individuals.

Regarding H2, discussing barrier gestures and social distancing measures separately could provide more insights. Our study showed that mindfulness was associated with higher compliance with social distancing measures, i.e., physical distancing and lockdown compliance. These two are effective preventive measures as they minimize the opportunities of being exposed to droplets and aerosols [87,88,89], although they are difficult to respect. The sacrifice of human contact and a habituation to COVID-19, which means that the anxiety and worry of the virus has diminished over time, are two of the obstacles that people face [90]. Being mindful means paying attention and flexibly adapting to the present environment, thus allowing people to be aware of the prevalence of the virus and better comply with social distancing and lockdown requirements. With respect to these two safety measures, we confirmed the results of other studies [61,62,63]. However, the more general statement that mindfulness leads to more compliance, as documented by [64] was not observed.

Indeed, the puzzling part related to barrier gestures: we only found one significant measure (i.e., coughing into your sleeve), while the MAAS score was positively correlated with the other three measures (i.e., washing hands, not touching faces and wearing masks), but only at the 10% significance level. We are standing at the critical point where we do not feel confident to declare or reject the predictive power of mindfulness. This result was in line with the study conducted by [61], in which they found no significant correlations between mindfulness and not touching faces, washing hands and disinfecting/cleaning frequently used surfaces. One possible supposition is that mindful people already limit their exposure to the virus through physical distancing and lockdown compliance, so mindfulness does not play a significant role in barrier gestures, which are more for better personal hygiene. Indeed, when an individual stays at home without contacting anyone, personal hygiene does not make a difference to the risk of infection. We regressed mindfulness onto the control variables and only age was positively correlated with the MAAS score. Consequently, these results suggested that the links between levels of mindfulness and compliance with government-proposed prophylactic measures were not homogeneous. The strong correlation between social distancing measures and mindfulness could offer opportunities for public policies to promote the practice of meditation during times of acute stress and violent exogenous shock for the general population in order to facilitate compliance with the required social distancing measures. On the other hand, the lack of a clear link to barrier gestures suggests that meditation practice could have limited impact on those measures. These findings that were obtained from a survey on a uniquely large sample are of interest because they are very difficult to observe in a laboratory and are theoretically hard to investigate. On the other hand, the size of the sample provided confidence in the robustness of the results.

Therefore, our study was a first step and new research should be conducted to include mindfulness with other factors that have been shown to be associated with compliant behaviors, such as empathy for others [91], belief in the effectiveness of health precautions [71], fear of COVID-19 [92] and political trust [93]. Further theoretical work should also investigate the underlying mechanisms that account for the differences in results across the prophylactic measures.

With all of the findings discussed above, we have accordingly contributed to the emerging literature that studies the buffering effects of mindfulness on well-being within the context of stressful situations and mindfulness as a determinant of compliance with preventive measures. The sample marked our study as the first to measure mindfulness by means of the MAAS scores of a national representative population in terms of gender, age and geographical area. In addition, our study revealed the general profile of the French population in a health crisis: how their well-being was impacted during COVID-19 and how they applied the prophylactic measures that were implemented by the government. What is more, we investigated the association between mindfulness and six prophylactic measures, while most other studies only focused on a single intervention.

Yet, our study had some limitations too. Firstly, more mindful people could also be more sensitive to the evolution of their sleep/mood. In the case of a degradation, mindful people could be more likely to report it. Were this true, our results would have a downward bias; thus, they should be considered as conservative. Moreover, we did not exogenously manipulate mindfulness in our survey, which only relied on cross-sectional data. Therefore, our results are not a genuine proof that mindfulness reduces negative impacts on well-being or increases the likelihood of respecting social distancing measures. Indeed, the causality could be reversed or produced by an unobserved variable. Secondly, self-report data, including the MAAS score, well-being status and compliance with six preventive measures might be biased by respondent attitude or self-estimation inaccuracy. Self-assessment and hypothetical biases refer to the tendency of humans to have and report biased knowledge about themselves [94]. For example, individuals usually overestimate their economic valuation of goods in hypothetical experiments [95,96] and overestimate their own abilities [97]. In particular, those biases are supposedly stronger when social desirability concerns are involved [98], which was the case in our study in which we asked participants about their degree of compliance with the prophylactic measures. Therefore, our dependent variables were likely to be overestimations of real behavior. Ultimately, our results were biased if the strength of the self-assessment bias was itself correlated with mindfulness. If mindful individuals were more biased in their self-assessment, then our study overestimated the real effects of mindfulness on actual behavior and well-being. In contrast, if mindful individuals were less prone to the self-assessment bias, as suggested by [99], then our results were indeed conservative and this study underestimated the real effects of mindfulness. In addition, a single item for measuring sleep quality and mood, respectively, could be too generic. Due to the emergency of COVID-19, we did not pretest our experimental material. However, since we analyzed compliance with each prophylactic measure and the impact of COVID-19 on mood and sleep quality separately instead of using a compliance and well-being index compounding several items, we believe that the reliability and validity of our measures were less of an issue. Lastly, there are some controversies surrounding the validity of the MAAS as a measure of mindfulness [100,101,102]. In particular, its ability to measure all aspects of mindfulness has been questioned and it is suspected to neglect facets such as the acceptance component [103] and non-judgmental awareness [104]. Nevertheless, there is a consensus that the MAAS measures one of the key aspects of mindfulness, at least: acting with awareness [105]. Therefore, it can be used as a proxy for mindfulness [101].

## 5. Conclusions

As discussed by [73,106], the emergence of COVID-19 has brought about new challenges for public policy from financial, organizational, social and behavioral perspectives. In a context of strained hospital resources, many governments around the world, are facing challenges in terms of management of diseases beyond COVID-19. Those include: the prevention, detection and treatment of psychological pathologies that result from the stress and threats that were generated by this pandemic; the modification of individual attitudes toward risk and ambiguity; and the difficulty for populations to respect barrier gestures and lockdowns that increase social isolation and potentially contribute to addictions. Despite some limitations to our study, our results support the idea that mindfulness could be an effective “inner hygiene”, which could help to maintain the health of individuals through better mood and sleep quality in the face of COVID-19. The practice of meditation could also promote compliance with some preventive measures and, consequently, reduce exposure to the virus. Public policies could promote the practice of meditation among the population to improve these two aspects at a relatively low cost. Indeed, nowadays, online courses and smartphone applications make mindfulness-based training easily accessible for the average person [107], which could enhance online interaction and foster a healthy mindset for dealing with stressful situations at the same time [106]. The main interventions, such as mindfulness-based stress reduction (MBSR) and acceptance and commitment therapy (ACT), can be feasibly and effectively adapted to online delivery [108,109,110] and, sometimes, can even be offered free of charge on well-known sites that are accessible to everyone, thereby avoiding social inequality. Our results were in line with [52], which point to the positive impact of meditation on the resilience of adolescents during the COVID-19 pandemic and emphasize that mindfulness training programs should be allowed more consideration in the future. Our results also support those of [53], which suggest that mindfulness meditation could be a viable low-cost response to mitigate the psychological burden of the COVID-19 crisis and any future pandemics. As a result, online mindfulness-based practices, along with personal hygiene and social distancing measures, should clearly be recommended by policy-makers as a complement to other policies, such as vaccination. Even during the post-COVID-19 period, mindfulness-based practices could be a low-cost approach that benefits the post-trauma recovery of the public.

## Figures and Tables

**Figure 1 ijerph-19-05051-f001:**
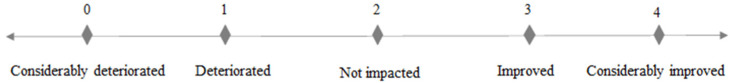
Integrated scale for impact on sleep quality and mood.

**Figure 2 ijerph-19-05051-f002:**
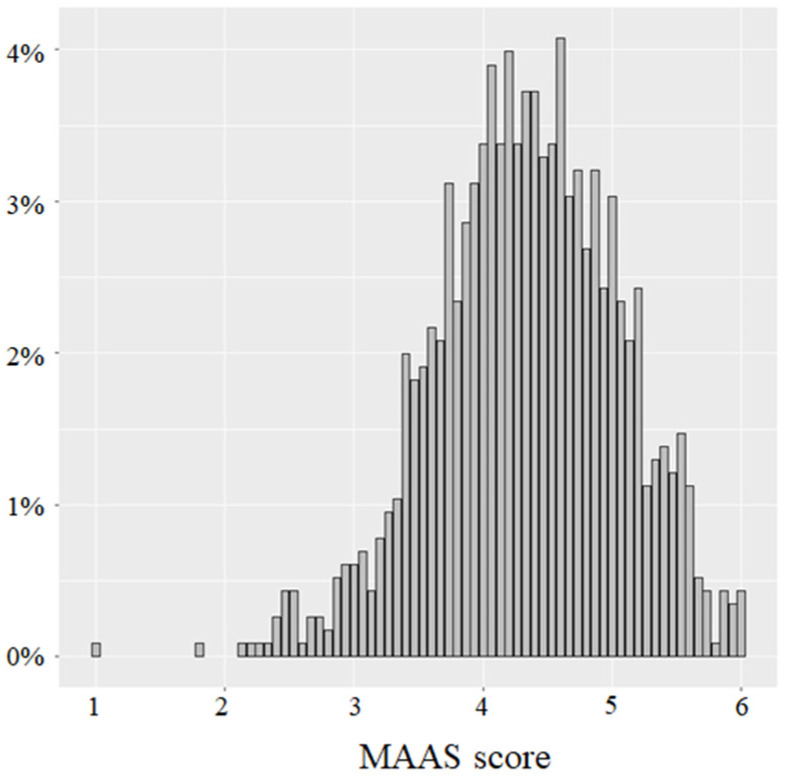
Distribution of the MAAS scores for our sample.

**Figure 3 ijerph-19-05051-f003:**
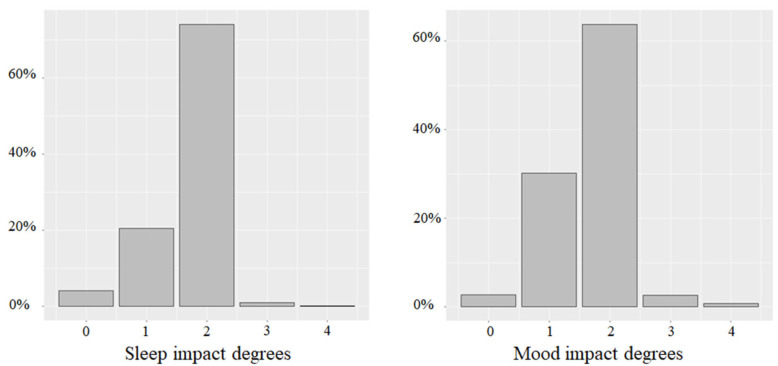
Distribution of impacts on sleep quality and mood impact. Note: 0 = “Considerably deteriorated”, 1 = “Deteriorated”, 2 = “No impact”, 3 = “Improved” and 4 = “Considerably improved”.

**Figure 4 ijerph-19-05051-f004:**
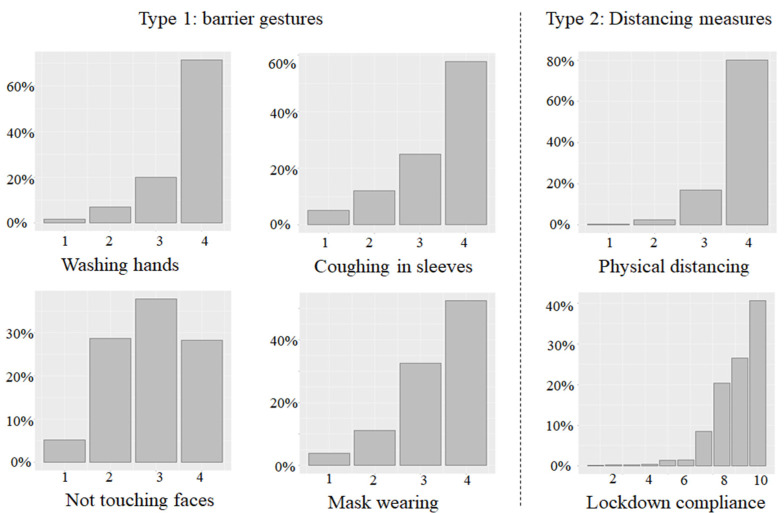
Compliance with each prophylactic measure. Note: For all measures except lockdown compliance: 1 = “Never”, 2 = “Sometimes”, 3 = “Often” and 4 = “Very often”. For lockdown compliance: degrees of compliance from 1 (“Not at all”) to 10 (“strictly”).

**Table 1 ijerph-19-05051-t001:** Representativeness of the sample.

	INSEE Census	Accepted (N = 5331)	Completed (N = 1154)
**Gender**			
Male	47.72%	47.77%	51.17%
Female	52.28%	52.23%	48.83%
**Age**			
[18, 24]	10.66%	8.59%	8.25%
[25, 34]	15.72%	15.52%	13.90%
[35, 49]	25.59%	25.44%	24.24%
[50, 64]	24.72%	27.45%	28.32%
[65, +∞]	23.31%	23.00%	25.28%
**Professional and Social Categories (INSEE definition: PCS)**			
Agriculteurs exploitants (Farmers)	0.94%	0.86%	0.78%
Artisans, commerçants, chefs entreprise (Craftsmen, merchants, business leaders)	3.33%	3.56%	4.17%
Cadres, professions intellectuelles sup, professions libérales (Executives, superior intellectual professions, liberal professions)	8.83%	9.01%	16.94%
Professions intermédiaires (Intermediate professions)	13.96%	15.57%	18.16%
Employés (Employees)	16.57%	16.87%	14.60%
Ouvriers (Workers)	13.38%	13.00%	7.73%
Retraités (Retired)	26.44%	28.24%	29.89%
Autres sans activité professionnelle (Others without professional activity)	16.55%	12.89%	7.73%
**Geographical Area (UDA-9 definition)**			
REGION PARISIENNE	18.79%	18.69%	17.29%
BP OUEST	9.31%	9.25%	8.34%
BP EST	7.76%	7.75%	7.73%
NORD	6.41%	6.34%	5.13%
OUEST	13.63%	13.51%	13.64%
EST	8.52%	8.52%	10.08%
SUD OUEST	10.94%	11.26%	12.16%
SUD EST	12.10%	12.12%	12.60%
MEDITERANEE	12.53%	12.57%	13.03%
**Urban Unit (INSEE definition)**			
<2000 inhabitants	22.51%	22.29%	21.98%
Between 2 k and 20 k	17.38%	17.67%	17.99%
Between 20 k and 100 k	13.55%	13.73%	13.55%
More than 100 k	29.88%	30.02%	31.28%
Parisian urban unit	16.67%	16.29%	15.20%

**Table 2 ijerph-19-05051-t002:** Definitions and descriptive statistics of the control variables.

Variable	Description	Mean (SD)	Median
Gender	1 = the individual reports being a male (NA = the individual reports “other”: removed for the analysis)	51.09% (0.50)	1
Age	In years	50.65 (16.98)	52
Monthly Income	Respondent’s monthly household income. 0 = “<EUR 1000”; 1 = “EUR 1000~2000”; 2 = “EUR 2000~3000”; 3 = “EUR 3000~4000”; 4 = “EUR 4000~5000”; 5 = “EUR 5000~6000”; 6 = “EUR 6000~7000”; 7 = “EUR 7000~8000”; 8 = “EUR 8000~9000”; 9 = “EUR 9000~10,000”; 10 = “EUR 10,000~15,000”; 11 = “>EUR 15,000” (NA = “I don’t know/I don’t want to reply”: removed from the analysis)	3.00 (2.28)	3
Education Level	0 = “No diploma”; 1 = “Certificate from primary school to high school”; 2 = “Bachelor’s degree”; 3 = “Master’s degree or PhD”	2.15 (0.78)	2
Vulnerable Person	1 = the respondent suffers a chronic illness that makes them vulnerable to the threat of COVID-19	44.89% (0.68)	0
Living Conditions	1 = the respondent lives with someone who is vulnerable to the threat of COVID-19 because of their age or a chronic illness	20.80% (0.41)	0

**Table 3 ijerph-19-05051-t003:** Comparison of the MAAS scores of general populations from different studies.

Study	Sample Size (Representative?)	Country	Mean	Standard Deviation	Cronbach α
[54] Brown and Ryan (2003)	74 (No)	USA	3.97	0.64	0.86
[75] Carlson and Brown (2005)	149 (No)	Canada	4.45	0.77	0.87
[76] Barajas and Garra (2014)	100 (No)	Spain	4.08	0.68	0.88
[77] Montes et al. (2014) ^1^	367 (No)	Argentina	3.88	-	0.87
Our sample	1154 (Yes)	France	4.33	0.71	0.84

^1^ There was no direct MAAS score in this study, but the mean score for each item was presented. We added the mean scores of all 15 items and then divided the sum by 15 to determine the mean that is listed here.

**Table 4 ijerph-19-05051-t004:** Pairwise correlations between the prophylactic measures.

	Lockdown Compliance	Mask Wearing	Coughing in Sleeves	Not Touching Faces	Physical Distancing
Washing hands	0.239 ***	0.159 ***	0.225 ***	0.216 ***	0.118 ***
Physical distancing	0.226 ***	0.211 ***	0.0512 **·**	0.215 ***	
Not touching faces	0.193 ***	0.212 ***	0.147 ***		
Coughing in sleeves	0.108 ***	0.125 ***			
Mask wearing	0.228 ***				

Note: *** Pearson correlation coefficient was at 0.1% level of significance; **·** Pearson correlation coefficient was at 10% level of significance.

**Table 5 ijerph-19-05051-t005:** Ordered probit regression models for well-being.

Ordered Probit Regression	(1)	(2)
Dependent Variable	Sleep Quality Impact Degree	Mood Impact Degree
MAAS score	0.1285 * (0.0568)	0.1516 ** (0.0525)
Men	0.3174 *** (0.0816)	0.1230 **·** (0.0744)
Age	0.0060 * (0.0026)	0.0103 *** (0.0023)
Monthly Income	0.0354 **·** (0.0190)	0.0278 (0.0172)
Education Level	−0.0833 (0.0562)	−0.0209 (0.0510)
Vulnerable Person	−0.1667 ** (0.0609)	−0.0593 (0.0565)
Living Conditions	−0.0374 (0.1002)	−0.0382 (0.0921)
/cut1	−0.9351 ** (0.2920)	−0.7524 ** (0.2707)
/cut2	0.1342 (0.2895)	0.7816 ** (0.2694)
/cut3	3.1413 *** (0.3124)	3.09371 *** (0.2824)
/cut4	3.8110 *** (0.3654)	3.6338 *** (0.2960)
N	1027	1027
Log-likelihood	−726.9622	−880.4104
AIC	1475.924	1782.821

Note: Standard error in parenthesis; **·** *p* < 0.1; * *p* < 0.05; ** *p* < 0.01; *** *p* < 0.001; 127 observations were removed as a consequence of providing no answer to the control questions.

**Table 6 ijerph-19-05051-t006:** Marginal effects of the MAAS scores on well-being.

Dependent Variable	Marginal Effect Type	Level 0 “Considerably Deteriorated”	Level 1 “Deteriorated”	Level 2 “No Impact”	Level 3 “Improved”	Level 4 “Considerably Improved”
Sleep Quality	MEM	−0.0099 * (0.0045)	−0.0290 * (0.0129)	0.0350 * (0.0155)	3.2506 × 10^−3^ * (1.6519 × 10^−3^)	6.9100 × 10^−4^ 5.4182 × 10^−4^
	AME	−0.0107 * (0.0049)	−0.0275 * (0.0122)	0.0338 * (0.0149)	3.6363 × 10^−3^ **·** (1.8597 × 10^−3^)	8.8847 × 10^−4^ **·** 6.7763 × 10^−4^
Mood	MEM	−0.0081 ** (0.0030)	−0.0460 ** (0.0161)	0.0430 ** (0.0151)	0.0076 ** (0.0029)	3.5500 × 10^−3^ * (1.5778 × 10^−3^)
	AME	−0.0089 ** (0.0033)	−0.0440 ** (0.0151)	0.0409 ** (0.0141)	0.0078 ** (0.0030)	0.0041 * (0.0018)

Note: Standard error in parenthesis; **·** *p* < 0.1; * *p* < 0.05; ** *p* < 0.01.

**Table 7 ijerph-19-05051-t007:** Ordered probit regression models for compliant behaviors.

Ordered Probit Models	(1)	(2)	(3)	(4)	(5)	(6)
Dependent Variable	Washing Hands	Coughing in Sleeves	Not Touching Faces	Mask Wearing	Physical Distancing	Lockdown
MAAS Score	0.1094 **·** (0.0582)	0.1518 ** (0.0538)	0.0922 **·** (0.0488)	0.0966 **·** (0.0525)	0.1356 * (0.0628)	0.1506 ** (0.0488)
Gender	−0.5998 *** (0.0841)	−0.2792 *** (0.0759)	−0.2555 *** (0.0695)	−0.3272 *** (0.0748)	−0.1513 (0.0910)	−0.3474 *** (0.0693)
Age	−0.0062 * (0.0025)	−0.0114 *** (0.0024)	0.0059 ** (0.0022)	0.0124 *** (0.0023)	0.0227 *** (0.0029)	0.0049 * (0.0021)
Monthly Income	0.0208 (0.0194)	0.0083 (0.0175)	0.0168 (0.0160)	−0.0034 (0.0168)	0.0167 (0.0218)	0.0191 (0.0160)
Education Level	−0.0216 (0.0546)	0.1145 * (0.0509)	0.0043 (0.0472)	0.0017 (0.0508)	0.0434 (0.0629)	0.0052 (0.0468)
Vulnerable Person	0.0174 (0.0625)	−0.0612 (0.0566)	−0.0146 (0.0528)	0.1785 ** (0.0580)	−0.0087 (0.0688)	0.0712 (0.0528)
Living Conditions	0.1296 (0.1031)	0.1036 (0.0934)	0.0052 (0.0863)	0.1650 **·** (0.0947)	0.0751 (0.1173)	0.1070 (0.0858)
/cut2	−2.3148 *** (0.3084)	−1.4880 *** (0.2790)	−1.0392 *** (0.2544)	−0.8406 ** (0.2726)	−0.9646 ** (0.3494)	−2.0832 *** (0.3307)
/cut3	−1.5240 *** (0.2967)	−0.7793 ** (0.2749)	0.1837 (0.2509)	−0.1126 (0.2685)	−0.2280 (0.3235)	−1.8556 *** (0.2973)
/cut4	−0.7030 * (0.2936)	0.0101 (0.2737)	1.2012 *** (0.2525)	0.9066 *** (0.2693)	0.9099 ** (0.3219)	−1.6636 *** (0.2808)
/cut5						−1.4964 *** (0.2710)
/cut6						−1.1633 *** (0.2594)
/cut7						−0.9306 *** (0.2550)
/cut8						−0.3251 (0.2503)
/cut9						0.4114 **·** (0.2495)
/cut10						1.1208 *** (0.2510)
N	1026	995	1007	1003	1025	1027
Log-likelihood	−804.4848	−1041.968	−1228.808	−1035.511	−569.0606	−1464.038
AIC	1628.97	2103.936	2477.617	2091.022	1158.121	2960.076

Note: Standard error in parenthesis; **·** *p* < 0.1; * *p* < 0.05; ** *p* < 0.01; *** *p* < 0.001. Different numbers of observations were removed as a consequence of providing no answer to the control questions and dependent variables.

**Table 8 ijerph-19-05051-t008:** Marginal effects of the MAAS scores on barrier gestures.

Dependent Variable	Marginal Effect Type	Level 1 “Never”	Level 2 “Sometimes”	Level 3 “Often”	Level 4 “Very Often”
Washing Hands	MEM	−3.7441 × 10^−3^ **·** (2.1203 × 10^−3^)	−0.0121 **·** (0.0065)	−0.0206 **·** (0.0110)	0.0364 **·** (0.0193)
	AME	−0.0045 **·** (0.0025)	−0.0121 **·** (0.0065)	−0.0185 **·** (0.0098)	0.0351 **·** (0.0186)
Coughing in Sleeves	MEM	−0.0145 ** (0.0053)	−0.0229 ** (0.0083)	−0.0220 ** (0.0081)	0.0595 ** (0.0211)
	AME	−0.0155 ** (0.0058)	−0.0218 ** (0.0078)	−0.0200 ** (0.0071)	0.0573 ** (0.0201)
Not Touching Faces	MEM	−0.0092 **·** (0.0049)	−0.0242 **·** (0.0128)	2.1515 × 10^−3^ (1.6241 × 10^−3^)	0.0312 **·** (0.0165)
	AME	−0.0094 **·** (0.0051)	−0.0235 **·** (0.0124)	2.0568 × 10^−3^ **·** (1.5472 × 10^−3^)	0.0309 **·** (0.0163)
Mask Wearing	MEM	−0.0072 **·** (0.0040)	−0.0138 **·** (0.0075)	−0.0174 **·** (0.0096)	0.0384 **·** (0.0209)
	AME	−0.0080 **·** (0.0045)	−0.0131 **·** (0.0072)	−0.0155 **·** (0.0084)	0.0366 **·** (0.0198)

Note: standard error in parenthesis; **·** *p* < 0.1; ** *p* < 0.01.

**Table 9 ijerph-19-05051-t009:** Marginal effects of the MAAS scores on physical distancing.

Dependent Variable	Marginal Effect Type	Level 1 “Never”	Level 2 “Sometimes”	Level 3 “Often”	Level 4 “Very Often”
Physical Distancing	MEM	−1.1141 × 10^−3^ (7.1219 × 10^−4^)	−0.0055 * (0.0027)	−0.0291 * (0.0135)	0.0357 * (0.0165)
	AME	−0.0018 (0.0011)	−0.0065 * (0.0032)	−0.0265 * (0.0123)	0.0349 * (0.0161)

Note: standard error in parenthesis; * *p* < 0.05.

**Table 10 ijerph-19-05051-t010:** Marginal effects of the MAAS scores on lockdown compliance.

Dependent Variable	Marginal Effect Type	Level 1~10 (From “Not at All” to “Strictly”)						
		1~4	5	6	7	8	9	10
Lockdown Compliance	MEM	n.s.	−4.0928 × 10^−3^ * (1.6652 × 10^−3^)	−4.3991 × 10^−3^ * (1.7395 × 10^−3^)	−0.0178 ** (0.0060)	−0.0244 ** (0.0081)	−3.3087 × 10^−3^ **·** (1.8140 × 10^−3^)	0.0579 ** (0.0187)
	AME	n.s.	−4.2632 × 10^−3^ (1.7452 × 10^−3^)	−4.4565 × 10^−3^ * (1.7648 × 10^−3^)	−0.0230 ** (0.0075)	−0.0174 ** (0.0058)	−3.0658 × 10^−3^ **·** (1.6639 × 10^−3^)	0.0565 ** (0.0182)

Note: standard error in parenthesis; **·** *p* < 0.1; * *p* < 0.05; ** *p* < 0.01; n.s., non-significant coefficient.

## Data Availability

Data were collected by the pool institute Viavoice and the Experimental Laboratory of Montpellier, University of Montpellier and are available upon request.

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
