# Peer review of "Did Mindful People Do Better during the COVID-19 Pandemic? Mindfulness Is Associated with Well-Being and Compliance with Prophylactic Measures"

_ijerph, 2022, doi:10.3390/ijerph19095051_

Round 1
Reviewer 1 Report
The submitted manuscript describes an investigation of the relations between mindfulness and a) sleep quality, b) mood, and c) compliance with COVID-19 prevention measures. They also test whether the relation between mindfulness and compliance with COVID-19 prevention measures is mediated by prosociability.
Introduction:
- The authors do a nice job of expanding their rationale for selecting each outcome.
- Hypotheses should be framed in the context of cross-sectional analyses, avoiding implying causality (e.g., H1: Mindfulness reduces adverse psychological effects of COVID-19, as captured by sleep quality and moods. ) The present analyses test associations between variables but cannot capture the effect of mindfulness in reducing any outcomes because no pre- to post- measures are collected.
Method:
- “Intuition” should not necessarily guide selection of covariates (p.8). Perhaps the authors meant “theory”, which combined with prior evidence is a good way to justify the selection of covariates for the present analyses. When the authors discuss rationale for these covariates, theory (with citations) may be used to explain observed patterns, but the use of intuition without citations or evidence for claims is not appropriate for a peer-reviewed publication.
Results:
- The comparison of MAAS scores with prior samples is helpful and a very interesting contribution to the literature. The authors could also consider including reliability for each.
Discussion:
- Causal language like “reduced” should be replaced in order to more accurately reflect the analyses and what they can tell us.
Reviewer 2 Report
I am very happy about the revised version of the paper: the authors have done a lot of work in correctly implementing all the comments of my previous report.
I only have one last general comment, as for the reference list at pp. 19-24. With respect to the previous version, the authors have exponentially enriched the survey of the literature, by adding almost 50 new references relevant to their study. I acknowledge that this might have taken a lot of time and effort. 
Maybe this is the reason why the way these references are presented in the reference list is not always consistent. For example:
- sometimes volume numbers are indicated with "Vol." before (e.g., reference no. 8), some other times without (e.g., reference no. 1);

- sometimes issue numbers are indicated with "n." before (e.g., reference no. 11), the overwhelming majority of other times without;

- sometimes page numbers are indicated with "pp." before (e.g., reference no. 1), some other times with "p." (e.g., reference no. 3), some other times without (e.g., reference no. 2);

- very few times you report the doi of the published article (e.g., reference no. 12); the majority of the other times you do not.
I would try to be consistent, e.g., volume, issue and page numbers respectively without "Vol.", "n." and "p." or "pp." before; or any other editing choice, as long as it is the same among all references. As for articles' doi, I would just do not report it for any article, since the other information provided are sufficient to find the articles.
Other specific typos I have detected in the first two pages of the reference list:
- Reference no. 4: "e1008274" is not the article page number, but its electronic number, so delete the "p." before.

- Reference no. 6. replace "Frontiers in Psychology, Vol. 12, n° November, p. 1–16" with "Frontiers in Psychology, 12, 703897".

- Reference no. 10: "104316" is not the article page number, but its number, so delete the "p." before.

- Reference no. 12: delete "ISSN 1389-9457" and add the article number "S1389-9457(21)00248-3".

- Reference no. 17: the article electronic number "e9" is missing.

- Reference no. 20: page numbering is not correct. Replace "822" with "822– 848".

- Reference no. 29: the name of the Journal, "Obesity Surgery", is repeated twice. Delete the repetition.

- Between Reference no. 29 and no. 30: "Chrisinger, B.W., Rich, T., Lounsbury, D., Peng, K., Zhang, J., Heaney, C.A., Lu, Y., & Hsing, A.W. (2021) [...]" should go to a new line/paragraph. This is the reason why it is not numbered (now it is between reference no. 29 and no. 30). It should be no. 30. Furthermore, delete "p." before "101451", which is the electronic number of the article.

- Reference no. 32: delete "p." before "1900", which is the electronic number of the article.

- Reference n. 35: delete "September", since you almost never report the month of the Journal issue throughout the reference list.

- Reference no. 37: I do not think that 50 is the page number. So, delete “p.”, or even “p. 50”, if “50” is the article number indicated within “4(50)”.

Please, fix similar typos (if any) also in the remaining four pages of the reference list (i.e., starting from reference no. 38).
Author Response
Dear reviewer,
Many thanks for all your remarks. We took into account all the points concerned.
Kind regards
The authors